# Sugar Composition of Thai Desserts and Their Impact on the Gut Microbiome in Healthy Volunteers: A Randomized Controlled Trial

**DOI:** 10.3390/nu16223933

**Published:** 2024-11-18

**Authors:** Sayamon Senaprom, Nuttaphat Namjud, Thunnicha Ondee, Akkarach Bumrungpert, Krit Pongpirul

**Affiliations:** 1Department of Preventive and Social Medicine, Faculty of Medicine, Chulalongkorn University, Bangkok 10330, Thailand; senaprom_s@hotmail.com (S.S.); ampere_nut@hotmail.com (N.N.); thunnichaon@yahoo.com (T.O.); 2Center of Excellence in Preventive and Integrative Medicine (CE-PIM), Faculty of Medicine, Chulalongkorn University, Bangkok 10330, Thailand; 3College of Integrative Medicine, Dhurakij Pundit University, Bangkok 10210, Thailand; abnutrition@yahoo.com; 4Bumrungrad International Hospital, Bangkok 10110, Thailand; 5Department of International Health, Johns Hopkins Bloomberg School of Public Health, Baltimore, MD 21205, USA; 6Department of Infection Biology & Microbiomes, Faculty of Health and Life Sciences, University of Liverpool, Liverpool L69 3GB, UK

**Keywords:** glycemic index, sugar composition, Thai desserts, gut microbiome profile

## Abstract

Background: The relationship between consuming Thai desserts—predominantly composed of carbohydrates—and gut microbiome profiles remains unclear. This study aimed to evaluate the effects of consuming various Thai desserts with different GI values on the gut microbiomes of healthy volunteers. Methods: This open-label, parallel randomized clinical trial involved 30 healthy individuals aged 18 to 45 years. Participants were randomly assigned to one of three groups: Phetchaburi’s Custard Cake (192 g, low-GI group, *n* = 10), Saraburi’s Curry Puff (98 g, medium-GI group, *n* = 10), and Lampang’s Crispy Rice Cracker (68 g, high-GI group, *n* = 10), each consumed alongside their standard breakfast. Fecal samples were collected at baseline and 24 h post-intervention for metagenomic analysis of gut microbiome profiles using 16S rRNA gene sequencing. Results: After 24 h, distinct trends in the relative abundance of various gut microbiota were observed among the dessert groups. In the high-GI dessert group, the abundance of *Collinsella* and *Bifidobacterium* decreased compared to the low- and medium-GI groups, while *Roseburia* and *Ruminococcus* showed slight increases. Correlation analysis revealed a significant negative relationship between sugar intake and *Lactobacillus* abundance in the medium- and high-GI groups, but not in the low-GI group. Additionally, a moderately negative association was observed between *Akkermansia* abundance and sugar intake in the high-GI group. These bacteria are implicated in energy metabolism and insulin regulation. LEfSe analysis identified Porphyromonadaceae and *Porphyromonas* as core microbiota in the low-GI group, whereas *Klebsiella* was enriched in the high-GI group, with no predominant bacteria identified in the medium-GI group. Conclusions: The findings suggest that Thai desserts with varying GI levels can influence specific gut bacteria, though these effects may be temporary.

## 1. Introduction

Diet is one of the main factors related to the diversity and dynamics of the gut microbiota [1]. Furthermore, dietary components have been found to influence the composition of the gut microbial community [2]. For example, bacteria in the genus *Bacteroides* are highly associated with the consumption of animal proteins and saturated fats, whereas bacteria in the genus *Prevotella* are associated with the consumption of carbohydrates and simple sugars, which are typical of agrarian societies [2,3]. There is concern that even brief changes in diet, especially a Westernized-style diet high in animal fat and sugar and low in plant-based fiber, may rapidly alter the composition and function of the gut microbiota. This has been observed in previous studies of animal models with elevated numbers of bile-tolerant, inflammation-associated Proteobacteria [4,5], and a decreased abundance of beneficial bacteria in the phylum Firmicutes [4,5,6]. The composition of the intestinal microbiota can be altered by long-term consumption of a habitual diet [2,7,8], as well as short-term consumption of specific diets [9]. Dietary changes may have temporary or permanent effects on the gut microbial profile [9,10]. For example, an acute change in diet, whether strictly animal-based or plant-based, alters the microbial composition within 24 h of initiation and returns to normal within 48 h after stopping the diet [9].

The previous study demonstrated an association between the consumption of digestible carbohydrates, such as monosaccharides and disaccharides, and an increase in pathogenic bacteria, as well as a decrease in species associated with the production of short-chain fatty acids (SCFAs) [11]. However, indigestible carbohydrates, or dietary fiber, are resistant to digestion in the small intestine and reach the large intestine [12], where they are fermented by bacteria, leading to increased SCFA levels and a beneficial effect on health [11]. A study by Ondee et al. [13] using an animal model found that there was a higher abundance of *Clostridium* bacteria in the consumption of high glucose and high fructose, while there was a lower abundance of *Allobaculum* in the high-fructose group compared to a regular diet. Furthermore, the administration of *Lactiplantibacillus plantarum* dfa1, which is a probiotic bacterium, increased the presence of *Lactobacilli* spp. (the beneficial bacteria) and the Chao1 index in mice fed glucose but not fructose compared to control mice fed a regular diet [13]. Santacruz et al. [14] reported a significant increase in *Bacteroides* abundance in mice fed a high-sugar diet. Another study found that the dietary pattern of consuming staple carbohydrate foods, including wheat, rice, and oats, for a consecutive week altered the structure of the intestinal microbial community, in which wheat and oats favored *Bifidobacterium catenulatum*, *B. bifidum*, and *B. adolescentis*, while rice suppressed *B. longum* and *B. adolescentis*, and wheat suppressed *Lactobacillus*, *Ruminococcus,* and *Bacteroides* [15]. Furthermore, Mano et al. [16] indicated that the abundance of bacteria in the phylum Actinobacteria and the genus *Bifidobacterium* was significantly higher after the intake of white bread compared to the intake of white rice.

A previous study using an animal model showed that the Bacteroidota phylum, specifically the *Bacteroides thetaiotaomicron* species, predominated the microbial community in a low-glycemic-diet group, while Firmicutes predominated in a high-GI-diet group [17]. An experimental study with individuals at risk for metabolic syndrome found that the genera *Bifidobacterium* and *Bacteroides* increased in the high-carbohydrate/high-GI-diet group compared to the control group, while the *Faecalibacterium prausnitzii* species increased in the high-carbohydrate/low-GI- and high-saturated-fat-diet groups [18].

In Thailand, the dietary habits are significantly different from those of Western diets. The Thai-style diet consists mainly of rice accompanied by a variety of dishes, such as meat, vegetables, noodles, and sometimes soup. Additionally, Thais often serve sweet desserts with their meals to counterbalance spicy dishes or as a snack [19]. In the past, Thais did not have desserts on a regular basis, as they do nowadays. The main ingredients of traditional Thai desserts consisted mainly of flour (such as rice flour or sticky rice flour), rice, palm sugar, and coconut milk [19,20]. Cultural exchanges with foreign countries have impacted the development of Thai desserts, introducing ingredients such as eggs, fat/butter, and dried beans. Refined sugar is now used in sweets instead of palm sugar, which is characteristic of traditional Thai desserts [19,20]. A study by Namjud et al. [21] demonstrated that Thai desserts exhibit a range of GI levels from low to high, including Phetchaburi’s Custard Cake, Saraburi’s Curry Puff, Nakhon Sawan’s Mochi, Suphan Buri’s Sponge Cake, Ayutthaya’s Cotton Candy, Prachuap Khiri Khan’s Pineapple Cheese Cake Biscuit, Chon Buri’s Bamboo Sticky Rice, and Lampang’s Crispy Rice Cracker. Several observational studies and clinical trials examined the relationship between snacks and the risk of metabolic disorders, finding that daily consumption of healthy snacks such as fruits, vegetables, nuts, and milk was associated with a reduced risk of metabolic syndrome. On the other hand, daily consumption of cookies, biscuits, and sweetened beverages was associated with a higher risk of metabolic syndrome [22,23,24,25].

The different glycemic indexes of desserts may influence the gut microbiota, but evidence is scarce. Therefore, this study aimed to determine the influence of the brief consumption of various Thai desserts with different GI levels on the gut microbiota. Desserts were added to a standard breakfast and the gut microbiome profiles of healthy participants were compared between dessert groups with low, medium, and high glycemic indexes.

## 2. Materials and Methods

### 2.1. Subjects

The protocol of this study was conducted in compliance with the Helsinki Declaration and Good Clinical Practice guidelines and approved by the Institutional Review Board of the Faculty of Medicine, Chulalongkorn University (IRB No. 0215/66, COA No. 0811/2023, approved on 22 June 2023). Recruitment and intervention occurred over a 3-month period from October to December 2023. A total of 96 healthy men and women volunteers from the project ‘Glycemic index and Glycemic load of local famous desserts in Thailand’ [21] were recruited for this study. At this step, 30 individuals declined to participate in the experiment. Following the signing of the informed consent form, the remaining 66 individuals were provided with questionnaires, which collected their information on basic characteristics such as sex, age, smoking, alcohol consumption, and habitual exercise. Subsequently, they were screened for eligibility and blood samples were collected for biochemical analysis and the measurement of anthropometrics and vital signs. The inclusion criteria were as follows: age 18–45 years, body mass index (BMI) in the range of 18.5–22.9 kg/m^2^, waist circumference in men ≤ 90 cm or women ≤ 80 cm, systolic blood pressure ≤ 120 mmHg, diastolic blood pressure ≤ 80 mmHg, and pulse 60–100 times per minute. People with fasting blood sugar < 100 mg/dL, hemoglobin A1c 4.2–6.0%, total cholesterol < 200 mg/dL, triglycerides < 150 mg/dL, high-density lipoprotein cholesterol (HDL-C) > 50 mg/dL for men or >40 mg/dL for women, low-density lipoprotein cholesterol (LDL-C) < 130 mg/dL, blood urea nitrogen 10–20 mg/dL, creatinine 0.6–1.2 mg/dL, alanine aminotransferase activity 0–48 IU/L, and aspartate aminotransferase activity < 35 IU/L were also included. The exclusion criteria were as follows: individuals who used antibiotics, prebiotics, or probiotic products within 1 month before enrollment; current smokers or smokers during the last 3 months; and individuals who drank alcoholic beverages and had gastrointestinal disease or diarrhea within 1 week before starting the study. At this step, another 30 individuals were excluded from the experiment. Finally, 36 participants were then screened as subjects in this study.

### 2.2. Study Design

The study was designed as an open-label, parallel randomized clinical trial in apparently healthy Thai adults (Registration number: TCTR20201008003). Before starting the study, the 36 participants were instructed on how to collect fecal samples and record dietary intake. Furthermore, they should prepare three days before the baseline visit and during the test periods by following the recommendations on dietary intake, including avoiding desserts, soft drinks, alcoholic beverages, yogurt, fermented foods (e.g., kimchi, pickles, miso, and kombucha), oligosaccharide-rich food (e.g., onions, leeks, garlic, asparagus, Jerusalem artichokes, and chicory root), and prebiotic and probiotic supplements. Additionally, they were advised to avoid physical exercise during the study period.

At the beginning of the study, at the baseline visit, the participants fasted overnight for 8–12 h before visiting. At the study center, their anthropometrics and vital signs were measured. Participants brought their fecal samples and dietary records. The fecal samples were preserved in 2 mL Eppendorf tubes containing PrimeStore MTM (a molecular transport medium that protects and stabilizes DNA and RNA) and transported on ice. The samples were stored frozen at −80 °C within 2 h after receipt until further analysis. The 36 participants were randomly assigned to one of three intervention groups (each group contained 12 participants) to consume three Thai desserts with varying levels of GI: group 1 received 192 g of Phetchaburi’s Custard Cake (low-GI dessert group), group 2 received 98 g of Saraburi’s Curry Puff (medium-GI dessert group), and group 3 received 68 g of Lampang’s Crispy Rice Cracker (high-GI dessert group). All desserts were portioned to provide 50 g of available carbohydrate content, and their nutritional values are described in the next section. Each dessert was weighed and purchased from the same production batch to control the nutrient composition. Participants consumed the Thai dessert portion provided in 15 min, with 150 mL of still water as the only beverage. After that, a standard breakfast meal was provided (grilled mackerel, Japanese style, with white rice; energy 538.8 kcal; carbohydrate 70.6 g; protein 20.1 g; fat 19.5 g). Subsequently, they were still provided with recommendations on the dietary intake for lunch and dinner. At the end of the study (after 24 h of intervention), the participants visited the study center again to bring their fecal samples and dietary records.

### 2.3. Thai Desserts for Testing

Thai desserts, also known as Khanom Thai, are traditional snacks. They are sweet foods that are integral to Thai cuisine, recognized for their delicious taste and elegant presentation. Due to their unique ingredients and preparation methods, they are considered part of the national identity [19,20]. Three Thai desserts with different GI values, Phetchaburi’s Custard Cake (low-GI dessert), Saraburi’s Curry Puff (medium-GI dessert), and Lampang’s Crispy Rice Cracker (high-GI dessert), were selected to study the gut microbiome profiles in the experiment [21]. To investigate the sugar composition of these desserts, three samples of each dessert were sent for analysis at the Food Quality Assurance Service Center (FQA), Institute of Food Research and Product Development (IFRPD), Kasetsart University, Thailand. The other nutrient composition data per 50 g of available carbohydrates (one serving size) of all desserts were retrieved from Namjud et al. [21], and are shown in Figure 1. More details of each dessert are explained below.

(1)Phetchaburi’s Custard Cake, also known as Khanom Maw Kaeng, is a well-known traditional Thai dessert that is a souvenir from the province of Phetchaburi. It is a sweet baked custard cake that consists mainly of eggs, coconut milk, sugar (palm sugar or refined sugar), and taro. It has a low glycemic index of 53 and contains 40.2 g of sugar, mostly sucrose (38.2 g).(2)Saraburi’s Curry Puff, also known as Karipap, is a crispy, deep-fried puff pastry with a clam-like shape stuffed with savory fillings such as chicken, potato, and onion. It is a famous souvenir dessert from Saraburi province. It has a medium glycemic index of 62 and contains 15.5 g of sugar, mostly sucrose (13.5 g).(3)Lampang’s Crispy Rice Cracker, also known as Nang Led or Khao Taen, is a deep-fried rice cracker drizzled with cane sugar. Made from sticky rice, watermelon, and cane sugar, it is a popular souvenir from Lampang province. It has a high glycemic index of 149 and contains 15.4 g of sugar, of which 10.9 g is sucrose.

The different levels of the GI dessert groups were classified based on the range of GI values as follows: low-GI dessert group with a GI value of ≤55; medium-GI dessert group with a GI value of 56–69; and high-GI dessert group with a GI value of ≥70 (using a scale in which pure glucose is given a GI value of 100) [27]. The serving sizes of the desserts were calculated by ensuring each provided 50 g of available carbohydrate content, as follows: (1) Phetchaburi’s Custard Cake had CHO 26.0 g of CHO per 100 g, so one serving size was (50 × 100/26.0) 192 g; (2) Saraburi’s Curry Puff contained CHO 51.1 g per 100 g, so one serving size was (50 × 100/51.1) 98 g; and (3) Lampang’s Crispy Rice Cracker had CHO 73.4 g per 100 g, so one serving size was (50 × 100/73.4) 68 g.

### 2.4. Measurement of Anthropometrics and Vital Signs

Anthropometric measurements, including body weight, height, waist circumference, pulse, and systolic and diastolic blood pressure, were measured and recorded during visits to the study center. Weight (to the nearest 0.1 kg) and height (to the nearest 0.1 cm) were measured early in the morning in the fasting state, with participants wearing light clothing without shoes. Based on weight and height, the BMI (kg/m^2^) value of each participant was calculated. Waist circumference was measured at the midpoint between the lower margin of the last palpable rib and the top of the iliac crest, using a tape measure. Systolic and diastolic blood pressure, as well as pulse, were measured in a sitting position.

### 2.5. Analysis of Blood Biochemical Parameters

Blood samples were taken from participants at the start of the study after an overnight fast of 8–12 h. Samples that would be used in various assays were collected into tubes containing different solutions as follows: sodium fluoride for the fasting blood sugar assay; EDTA for the hemoglobin A1c assay; and clotted blood for the measurement of lipid profiles (total cholesterol, LDL-C (direct), HDL-C, triglycerides), creatinine, blood urea nitrogen, alanine aminotransferase, and aspartate aminotransferase activity. Subsequently, the samples were centrifuged at 3000× *g* rpm for 20 min at 4 °C, and then the collected plasma and serum samples were stored at −80 °C until analysis. All biochemical analyses were performed by the HIV-NAT Research Laboratory, Faculty of Medicine, Chulalongkorn University, Thailand.

### 2.6. Dietary Intake Assessment

Briefly, the participants were given both written and verbal instructions by trained fieldworkers on the purpose of the dietary record and how to record food items consumed in the forms. Participants provided data on their food intake for 3 days before the start of the study and during the 24 h test period to determine their energy and macronutrient intakes using the food record forms. Energy and macronutrient intake were estimated using the INMUCAL nutrients software (Version 4.0) of the Institute of Nutrition, Mahidol University in Thailand [28].

### 2.7. DNA Extraction and 16S rRNA Gene Next-Generation Sequencing

Microbial DNA was extracted from 0.25 g of each frozen fecal sample using the QIAamp PowerFecal Pro DNA Kit (Qiagen, Hilden, Germany) according to the manufacturer’s protocols. The quantity of extracted DNA samples was determined using the DeNovix QFX Fluorometer (DeNovix Inc., Wilmington, DE, USA). The QIAseq 16S/ITS Region Panels (Qiagen, Hilden, Germany) were used to prepare the amplified DNA libraries targeting the V3V4 region of the bacterial 16S rRNA gene and label the amplicons with different sequencing adaptors. The quality and quantity of libraries were evaluated using the QIAxcel Advanced System (Qiagen, Hilden, Germany) and the DeNovix QFX Fluorometer, respectively. Finally, the paired-end sequencing of read lengths of 2 × 300 bp was conducted by Illumina MiSeq System using the MiSeq Reagent Kit v3 (600-cycle) (Illumina, San Diego, CA, USA).

### 2.8. Bioinformatic Analysis

The paired-end sequence reads in FASTQ format were processed using the DADA2 v1.16.0 pipeline [29] (available at https://benjjneb.github.io/dada2/, accessed on 1 October 2023). The DADA2 pipeline describes microbial diversity and community structures using unique amplicon sequence variants (ASVs). Microbial taxa were classified based on 16S rRNA gene sequences using SILVA SSU version 138 as a reference database [30]. Alpha diversity analyses refer to the bacterial diversity within each community [31]. Alpha diversity indices, including observed ASVs, Chao1, Shannon, and phylogenetic diversity (PD) whole tree, were computed using the DADA2 pipeline. Beta diversity analyses represent the difference in microbial community between samples. To estimate the beta diversity, non-metric multidimensional scaling (NMDS) based on Bray–Curtis dissimilarity and principal coordinate analysis (PCoA) on weighted/unweighted UniFrac distances and generalized UniFrac (GUniFrac) distances were plotted using the program Phyloseq [32] (available at https://joey711.github.io/phyloseq/, accessed on 1 October 2023). Taxonomic relative abundance profiles at taxa levels (phylum, class, order, family, and genus) were generated based on ASV annotation. Finally, the linear discriminant analysis effect size (LEfSe), which was usually used to determine the significantly higher taxonomies, genes, or functions and explain the difference in taxa between groups [33], was performed to identify the bacterial biomarkers in each dessert group.

### 2.9. Statistical Data Analysis

Categorical data were reported as frequency and percentage, and were analyzed by the Chi-squared test or Fisher’s exact test. Descriptive data were presented as mean ± standard deviation (SD) or median (25th and 75th percentile) for continuous variables. Microbial taxa in relative abundance (%), indices of taxonomic alpha diversity, and continuous variables with normal distribution were analyzed using one-way analysis of variance (One-Way ANOVA), followed by Bonferroni post hoc test for multiple pairwise comparisons. For non-normally distributed continuous variables, the Kruskal–Wallis test was used, followed by Dunn’s post hoc test. The *p*-values were adjusted for multiple hypothesis testing using the Benjamini–Hochberg procedure, and the significance threshold was set at adjusted *p* < 0.05. The Kruskal–Wallis test was also used in LEfSe analysis with a linear discriminant analysis (LDA) score ≥ 1.0 to find biomarkers for bacteria that differed significantly in abundance between sample groups. The permutational multivariate analysis of variance (PERMANOVA) was performed to evaluate the significant differences in beta diversity between groups at *p* < 0.05. The associations between intakes of carbohydrates, sugars, protein, and fat, and relative abundances of bacteria were investigated by Spearman’s rank correlation coefficient test. Statistical analyses were conducted as two-sided. Statistically significant differences and tendencies were considered at *p* < 0.05 and 0.05 ≤ *p* < 0.10, respectively. All statistical analyses were performed using Stata software version 18.0 (STATA Corp., College Station, TX, USA).

## 3. Results

### 3.1. Characteristics of Participants

The design of this study followed the Consolidated Standards of Reporting Trials (CONSORT) reporting guideline, as shown in Figure 2. From the total of 36 participants in this experiment, 1 participant was infected with COVID-19 before the start of the study, 2 participants failed to follow the protocol during the testing period, and 3 participants were unable to collect fecal samples at the baseline and after 24 h intervention. Consequently, 30 participants, with 10 participants in each testing dessert group, were included in the study analysis. The characteristics of the participants are reported in Table 1. The average age of the enrolled participants in the low-GI, medium-GI, and high-GI dessert groups was 29.2 ± 7.3 years, 30.7 ± 7.8 years, and 28.2 ± 7.2 years, respectively. The average BMI of participants in the low-GI, medium-GI, and high-GI dessert groups was 20.8 ± 1.4 kg/m^2^, 20.9 ± 1.0 kg/m^2^, and 20.3 ± 1.5 kg/m^2^, respectively. Furthermore, there were no significant differences in gender, anthropometrics, vital signs, blood biochemical parameters, or duration of fecal sample collection between groups as statistically analyzed using Fisher’s exact test for categorical data, or One-Way ANOVA or the Kruskal–Wallis test for continuous data.

### 3.2. Energy Intake and Dietary Composition

The energy intake and dietary composition of the participants are reported in Table 2. At baseline, the average daily energy intake in the low-GI, medium-GI, and high-GI dessert groups was 1172.9 ± 252.4 kcal/day, 1151.3 ± 250.8 kcal/day, and 1049.3 ± 232.8 kcal/day, respectively. After the 24 h intervention, the energy intake in the low-GI, medium-GI, and high-GI dessert groups was 1425 ± 457.5 kcal/day, 1302 ± 337.3 kcal/day, and 1503.6 ± 378.4 kcal/day, respectively. By statistical analysis with One-Way ANOVA or the Kruskal–Wallis test, there were no significant differences in energy intake and dietary composition between the testing dessert groups, both at the baseline and after 24 h intervention, in terms of total energy intake, macronutrients, dietary fiber, and the energy intake from carbohydrates, protein, and fat. However, total sugar intake increased during the 24 h intervention in all testing dessert groups, resulting in significant differences in the low-GI dessert group [56.9 (43.5, 62.2) g/day] compared to the medium-GI dessert group [25.6 (20.8, 41.8) g/day] and the high-GI dessert group [27.7 (22.9, 33.8) g/day].

### 3.3. Gut Microbiota Composition

The gut microbiome of 30 participants in all three dessert groups was profiled through a total of 6,243,685 paired-end reads obtained from 16S rRNA gene next-generation sequencing. The raw reads were subsequently processed by trimming adapters, filtering out low-quality reads, merging paired-end reads into consensus sequences, screening for non-chimeric reads, and removing mitochondrial 16S rRNA. At baseline, 103,607.3 ± 20,314.5 reads were obtained from the low-GI dessert group, 96,832.2 ± 18,274.6 reads for the medium-GI dessert group, and 87,621.4 ± 14,144.6 reads for the high-GI dessert group. After the 24 h intervention, the number of reads in these groups was 116,696.2 ± 15,660.9, 120,812.6 ± 30,058.8, and 98,798.8 ± 13,159.5, respectively. There were no significant differences in sequence read numbers between all dessert groups, both at the baseline (*p* = 0.150) or after the 24 h intervention (*p* = 0.076). The bacterial taxonomic identification based on 16S rRNA gene sequences revealed that the gut microbiota in participants was composed of 11 phyla, 16 classes, 37 orders, 80 families, and 227 genera (Appendix A). However, when focusing on only the top 100 taxa, merely five phyla, eight classes, 14 orders, 20 families, and 41 genera dominated the gut microbiome profile, as shown in Figure 3.

At the phylum level (Appendix A), bacteria in the phylum Firmicutes and Bacteroidota dominated the gut microbial ecosystem, with Actinobacteriota, Proteobacteria, and Fusobacteriota as minor components. The relative abundance of Firmicutes at both the baseline and after the 24 h intervention was >60%. At the class level (Appendix A), across all dessert groups, the most abundant class was Clostridia with a relative abundance of >50%, followed by Bacteroidia, Actinobacteria, and Bacilli, respectively. At the order level (Appendix A), the most abundant bacteria were in the order Lachnospirales, followed by Bacteroidales, Oscillospirales, and Bifidobacteriales, respectively. The most abundant family was Lachnospiraceae, followed by Bacteroidaceae, Ruminococcaceae, Bifidobacteriaceae, and Prevotellaceae, respectively (Appendix A). With more than two hundred genera, the microbiota composition at the genus level showed that *Blautia* and *Bacteroides* were the two most abundant genera, each with a relative abundance of >10% (Appendix A). The other common genera included *Bifidobacterium*, *Faecalibacterium*, *Prevotella*, *Agathobacter*, *Collinsella*, *Dorea*, *Roseburia*, *Coprococcus*, *Fusicatenibacter*, *Holdemanella*, *Subdoligranulum*, *Anaerostipes*, *Lachnoclostridium*, *Ruminococcus*, and *Streptococcus*.

### 3.4. Changes in Gut Microbiota After 24 h Consumption of Thai Desserts

The changes in relative abundance of all identified taxa after consuming three Thai desserts were calculated with the following formula: the relative abundance (%) at the 24 h intervention minus the relative abundance (%) at the baseline for each participant and identified taxon. Subsequently, the changes in the relative abundance of some dominant phyla, classes, orders, families, and genera in each dessert group were statistically analyzed and represented as boxplots, as shown in Figure 4. At the phylum level (Figure 4A), the median change in Bacteroidota was slightly increased in the high-GI dessert group compared to the medium-GI dessert group. However, the median change in Actinobacteriota was significantly decreased in the high-GI dessert group compared to the low-GI and medium-GI dessert groups. The median change in Actinobacteria at the class level (Figure 4B) was slightly decreased in the high-GI dessert group compared to the low-GI and medium-GI dessert groups. On the other hand, the median change in Coriobacteriia was significantly different between the groups, with an increase in the low-GI and medium-GI dessert groups compared to the high-GI dessert group. In the order Erysipelotrichales (Figure 4C), the median change in the high-GI dessert group showed a significant decrease compared to the low-GI dessert group, but a slight decrease in the medium-GI dessert group. Similar results were also observed in Bifidobacteriales, in which the median change in the high-GI dessert group decreased when compared to the low-GI and medium-GI dessert groups, however, both without significance. At the family level (Figure 4D), the median change in Coriobacteriaceae in the low-GI and medium-GI dessert groups increased significantly compared to the high-GI dessert group. In Erysipelotrichaceae, the high-GI dessert group had a significant decrease in median change compared to low-GI and medium-GI dessert groups. On the other hand, in Peptostreptococcaceae, the high-GI dessert group showed a significant increase in the median change compared to the low-GI dessert group.

At the genus level (Figure 4E), changes in the relative abundance of a number of bacterial genera were found to be interesting. The genera *Holdemanella*, *Roseburia*, *Eubacterium hallii*, *Fusicatenibacter*, *Ruminococcus*, and *Romboutsia* belong to the phylum Firmicutes, while *Bifidobacterium* and *Collinsella* belong to the phylum Actinobacteriota. The median change in the high-GI dessert group in the genus *Holdemanella* decreased significantly compared to the low-GI dessert group, but slightly decreased compared to the medium-GI dessert group. In *Roseburia,* the high-GI dessert group showed a slight increase in median change compared to the low-GI dessert group. The median change in the *Eubacterium hallii* levels of the low-GI dessert group increased slightly compared to the medium-GI dessert group, but increased significantly compared to the high-GI dessert group. Similar changes in abundance were also observed in *Fusicatenibacter*, in which the low-GI dessert group had a significant increase compared to both the medium-GI and high-GI dessert groups. The median change in the high-GI dessert group increased slightly compared to the medium-GI dessert group in the genus *Ruminococcus*. In addition, in *Romboutsia*, the high-GI dessert group showed a significant increase in median change compared to the low-GI dessert group. For the two genera in another phylum, the median change in both *Bifidobacterium* and *Collinsella* of the low-GI and medium-GI dessert groups increased compared to the high-GI dessert group, not significantly in the former genus but significantly in the latter.

When focusing on only the relevant changes in relative abundance in the high-GI dessert group, the bacteria in the phylum Actinobacteriota, classes Actinobacteria and Coriobacteriia, orders Erysipelotrichales and Bifidobacteriales, family Coriobacteriaceae, and genera *Collinsella, Bifidobacterium*, *Holdemanella*, *Eubacterium hallii*, and *Fusicatenibacter* showed slightly or significantly lower abundance compared to the low-GI and medium-GI dessert groups. On the other hand, the bacteria in the phylum Bacteroidota, family Peptostreptococcaceae, and genera *Roseburia*, *Ruminococcus*, and *Romboutsia* had a slight or significant increase in abundance when compared to those two dessert groups.

### 3.5. Biomarkers for Gut Microbiota Associated with Consumption of Thai Desserts

To confirm and identify gut microbial alterations associated with the consumption of Thai desserts at the taxonomic level, the LEfSe analysis was used to identify the taxa that most influenced observations as shown in Figure 5. Only the bacterial taxa with LDA scores (absolute value) of ≥1 with statistically significant differences would be defined as dominant biomarkers. The LEfSe results showed that only two dessert groups, low-GI and high-GI, were determined for the dominant biomarkers, while the medium-GI dessert group did not have significant differences in the gut microbiome profile. After consumption of the low-GI dessert, bacteria in the family Porphyromonadaceae and the genus *Porphyromonas*, which belong to the phylum Bacteroidota, were found as the core gut microbiota. Moreover, in the high-GI dessert group, bacteria in the genus *Klebsiella* were discovered to be highly prevalent. These results indicated that after the consumption of low-GI and high-GI desserts for 24 h, there were actually changes in the gut microbiome profiles of participants consuming these two desserts.

### 3.6. Diversity of Gut Microbiome Profiles

The alpha and beta diversity indices were analyzed to measure the diversity of the gut microbiota after 24 h of consumption of three desserts. The alpha diversity refers to the intra-individual diversity or diversity of a single sample group, while the beta diversity refers to the inter-individual diversity and would be used to estimate the similarity or dissimilarity between the sample groups. For alpha diversity, four indices including observed ASVs, bacterial richness with Chao1, bacterial evenness with Shannon, and PD whole tree were used to measure the diversity of the gut microbiome of 10 participants at the baseline and after the 24 h intervention in the low-GI, medium-GI, and high-GI dessert groups. The results of the analysis showed that there were no significant differences between the low-GI, medium-GI, and high-GI dessert groups in any alpha diversity indices at the beginning and after the test (Figure 6), which indicated that there were no differences in the diversity of the gut microbiome among the participants consuming all desserts. These similarities were also supported by the beta diversity indices including PCoA on weighted and unweighted UniFrac distances, GUniFrac distances, and NMDS based on Bray–Curtis dissimilarity. The statistical analysis by the PERMANOVA test showed that the gut microbiota composition at baseline and after the 24 h intervention was very close in all dessert groups, with no significant difference in any beta diversity indices, as shown in Figure 7. The findings of alpha and beta diversity analyses hence suggested that the gut microbiome profiles of participants at baseline and 24 h after consumption of all desserts were still similar in bacterial structure.

### 3.7. Correlation Between Dietary Nutrients and Relative Abundances of Gut Microbiota

To determine the relationship between dietary nutrient intake and gut microbiome profiles, Spearman’s rank correlation coefficient was used to calculate the strength and direction of association, or Spearman’s rho (*ρ*), between dietary nutrient intake and the relative abundance of each bacterial genus, as shown in Figure 8. In the low-GI dessert group, there were significantly positive relationships between the genus *Lachnoclostridium* and the intake amount of carbohydrates (*ρ* = 0.90, *p* < 0.001), sugar (*ρ* = 0.88, *p* < 0.001), and fat (*ρ* = 0.68, *p* = 0.029). The genus *Collinsella* had a significantly positive association with protein intake (*ρ* = 0.70, *p* = 0.025), while *Faecalibacterium* demonstrated a significantly positive relationship with fat intake (*ρ* = 0.73, *p* = 0.016). However, there were significantly negative correlations between the intake of carbohydrates and sugar and the abundance of bacteria in *Clostridium sensu stricto 1* (*ρ* = −0.75, *p* = 0.013 and *ρ* = −0.68, *p* = 0.029), *Dorea* (*ρ* = −0.70, *p* = 0.028 and *ρ* = −0.64, *p* = 0.047), and *Romboutsia* (*ρ* = −0.79, *p* = 0.009 and *ρ* = −0.67, *p* = 0.036). Furthermore, the relative abundance of *Bacteroides* (*ρ* = −0.83, *p* = 0.005) and *Fusobacterium* (*ρ* = −0.81, *p* = 0.005) showed a significantly negative correlation with protein intake in this dessert group.

For the medium-GI dessert group, the genus *Blautia* had a significantly positive relationship with carb intake (*ρ* = 0.66, *p* = 0.040), and *Coprococcus* had a significantly positive relationship with both carb and fat intake (*ρ* = 0.84, *p* = 0.004, and *ρ* = 0.68, *p* = 0.029, respectively). On the other hand, there was a significantly negative correlation between sugar intake and the relative abundance of *Lachnospiraceae UCG-010* (*ρ* = −0.66, *p* = 0.040), while *Agathobacter* had a significantly negative relationship with protein intake (*ρ* = −0.68, *p =* 0.029), and *Lactobacillus* showed a significantly negative correlation with sugar intake (*ρ* = −0.68, *p* = 0.033).

In the high-GI dessert group, the relative abundance of the *Lactococcus* genus had a significantly positive relationship with carbohydrate intake (*ρ* = 0.78, *p* = 0.010), while *Lachnospira* had a significantly positive correlation with sugar intake (*ρ* = 0.66, *p* = 0.038). Moreover, *Enterococcus* showed a significantly positive correlation with the intake of carbohydrates (*ρ* = 0.78, *p =* 0.010), protein (*ρ* = 0.66, *p =* 0.040), and fat (*ρ* = 0.65, *p =* 0.045). On the other hand, the abundance of *Megamonas* showed a significantly negative correlation with carbohydrate intake (*ρ* = −0.66, *p =* 0.039), while *Lactobacillus* had a significantly negative relationship with sugar intake (*ρ* = −0.64, *p* = 0.049).

In addition to strong and significant correlations, there was also a tendency for correlation in some relationships. The genus *Desulfovibrio* showed a tendency for a negative correlation with the amount of fat intake (*ρ* = −0.59, *p* = 0.076) in the low-GI dessert group, while *Lactococcus* had a tendency for a negative relationship with carbohydrate intake (*ρ* = −0.62, *p* = 0.056) in the medium-GI dessert group. In the high-GI dessert group, there was a tendency toward a negative relationship between the abundance of *Lachnoclostridium* and carbohydrate intake (*ρ* = −0.56, *p* = 0.090) and between *Akkermansia* and sugar intake (*ρ* = −0.52, *p* = 0.090). On the other hand, *Lactococcus* showed a tendency of positive correlation with sugar intake (*ρ* = 0.61, *p* = 0.062).

## 4. Discussion

In the present study, we examined the influence of consuming three Thai desserts with different GI levels—Phetchaburi’s Custard Cake (low-GI), Saraburi’s Curry Puff (medium-GI), and Lampang’s Crispy Rice Cracker (high-GI)—on the gut microbiome profiles of healthy volunteers over a brief 24 h period. According to our results, the dominant phyla with high relative abundance were Firmicutes (65.9%), Bacteroidota (20.7%), Actinobacteria (8.8%), and Proteobacteria (3.0%), with subdominant phyla including Fusobacteria (1.1%), Desulfobacterota (0.3%), Verrucomicrobiota (0.1%), and Campilobacterota (0.01%). These findings align with those reported by Donaldson et al. [34], Guarner and Malagelada [35], and Hollister et al. [36]. In another study, Firmicutes and Bacteroidota accounted for more than 70% of total bacteria [37], playing essential roles within the gut ecosystem. Our findings for Firmicutes and Actinobacteria are consistent with La-ongkham et al. [38] in their assessment of the main taxonomic features of the Thai gut microbiome. However, compared to La-ongkham et al., we observed higher Bacteroidota and lower Proteobacteria, with our results on Proteobacteria aligning with those of Raethong et al. [39]. Individual variations in microbiota proportions can lead to functional differences associated with host health [40]. Additionally, studies have shown that gut microbiota composition can shift within a day when transitioning between diets with different macronutrient compositions [2,6,41].

In our study, the high-GI dessert group showed a slight decrease in Firmicutes and a slight increase in Bacterioidota (Appendix A). The Firmicutes phylum includes many important Gram-positive bacteria, with some, such as *Lactobacillus* and *Lactococcus*, playing roles in fermentation and bacteriocin production [37,42]. The ingredients in the Thai desserts likely influenced these changes: Phetchaburi’s Custard Cake (low-GI) consists of eggs, coconut milk, palm/refined sugar, and taro; Saraburi’s Curry Puff (medium-GI) contains wheat flour, chicken, potato, and onion; and Lampang’s Crispy Rice Cracker (high-GI) is primarily sticky rice, watermelon, and cane sugar. All participants consumed white rice as a staple and 10% also consumed glutinous rice (Phetchaburi’s Custard Cake *n* = 0, Saraburi’s Curry Puff *n* = 2, and Lampang’s Crispy Rice Cracker *n* = 1). Dietary habits impact gut microbiota, with studies like Zhang et al.’s [43] suggesting that diet accounts for 60% of gut microbial composition. Phoonlapdacha et al. [44] also associated glutinous rice with Bacteroidota abundance in pregnant Thai women, likely due to its low amylose and high amylopectin content [45], which contributes to the unique nutritional qualities of processed foods [46]. In in vitro studies, amylopectin was found to be more fermentable than resistant starch [47]. Studies using animal models showed Fermicutes to be the predominant phylum in high-GI diets, while Bacteroidota (e.g., *Bacteroides thetaiotaomicron*) predominated in low-GI diets [17]. Another study found that the prevalence of Firmicutes increased, and the Bacteroidota population reduced in the high-sugar-diet group in an animal model [48].

It is well known that carbohydrate types impact blood glucose levels, with different digestion rates affecting blood availability and fermentation potential in the colon [49]. Furthermore, the consumption of different types and amounts of carbohydrates may also have effects on the gut microbiome [17,18]. Western diets tend to be high in fat and sugar but low in plant-based fiber. Many high-carbohydrate foods that are common in Western diets, such as refined cereals, corn, potatoes, and sugars (sucrose and fructose), have a high glycemic response (or what are called high-GI foods) [50,51]. Consumption of excessive amounts of carbohydrates, especially those high in glucose and fructose, may lead to metabolic disorders and gut dysbiosis [52]. Additionally, it has been associated with an increased presence of pathogenic bacteria, specifically *Bacteroides* [53]. One study, supportably, found that the *Bacteroides* enterotype was associated with significant meat consumption, while the *Prevotella* enterotype was associated with high carbohydrate consumption, particularly sugar [54]. A reduced abundance of beneficial bacteria, such as *Faecalibacterium*, *Roseburia*, and *Ruminococcus*, which produce short-chain fatty acids, was found in Western diets [8,55]. In our study, *Roseburia* and *Ruminococcus* showed a slight increase in the high-GI dessert group, while *Faecalibacterium* remained stable across groups (Appendix A). Correlation analysis indicated a strong negative association between sugar intake and *Lactobacillus* in the medium- and high-GI dessert groups, but not in the low-GI group. We also observed a tendency for a moderately negative relationship between *Akkermansia* abundance and sugar intake in the high-GI group. *Lactobacillus*, *Bifidobacterium*, *Clostridium*, and *Akkermansia* can influence insulin resistance and inflammation by modulating carbohydrate absorption and energy use [56].

We observed a strong negative correlation between carbohydrates and sugar intake and the abundance of *Clostridium sensu stricto 1*, *Dorea*, and *Romboutsia*. In addition, there was a strong negative correlation between protein intake and the relative abundance of *Bacteroides* and *Fusobacterium* in the low-GI group. Additionally, the analysis showed that there was a moderate negative relationship between the abundance of *Desulfovibrio* and fat intake in the low-GI dessert group. However, Seel et al. [57] found that people who ate Western diets that had several high-carbohydrate foods had the highest levels of *Dorea*, *Eubacterium ruminantium group*, *Ruminococcus torques group*, Ruminococcaceae, Lachnospiraceae, *Lactobacillus*, and *Senegalimassilia*. A study conducted with animals fed a high-carbohydrate diet showed that there was an increase in the ratio of bacteria in the Firmicutes phylum to Bacteroidota, as well as an increase in pro-inflammatory *Desulfovibrio vulgaris* and mucin-degrading *Akkermansia muciniphila* [58]. In this study, nutrient intakes, which were the average of total energy intake, macronutrients, dietary fiber, and the energy intake from carbohydrate, protein, and fat, did not differ between dessert groups, whereas sugar intake in the low-GI dessert group was higher than in the medium-GI and high-GI dessert groups during the intervention. Diet can influence the composition and richness of the gut microbiota. As one of the macronutrients, carbohydrates are likely to play an important role in this process [9,59,60]. Nonglycemic carbohydrates, like dietary fiber, are not hydrolyzed by human enzymes and may be fermented in the large intestine. On the other hand, glycemic carbohydrates, like sugars (except alcoholic sugars) and starches (wheat, rice, maize, and potato), are hydrolyzed to monosaccharides; for example, 5–30% of fructose is absorbed in the small intestine, but glucose is absorbed actively [61]. Due to different absorption mechanisms between the two monosaccharides occurring in the small intestine, fructose molecules are absorbed at a slower rate than glucose molecules. It seems that all of these sugars are substrates for microorganisms in both the small and large intestine [62]. The effects of sugars on the physiology of gut microbes and their influence on the disruptive balance between beneficial and non-beneficial gut microbiota were discussed in a previous study [63]. Additionally, Phetchaburi’s Custard Cake, which is the low-GI dessert, had the highest sugar content in the nutrient test (40.2 g/carbohydrate 50 g), which was mostly sucrose (38.2 g/carbohydrate 50 g), when compared to the high-GI dessert, Lampang’s Crispy Rice Cracker (sugar 15.4 g/carbohydrate 50 g and sucrose 10.9 g/carbohydrate 50 g). However, total sugar intake from the diet in the low-GI dessert group (55.6 g) was higher than the high-GI dessert group (36.9 g), while the total carbohydrate and dietary fiber intake from the diet were not different between these dessert groups. Saraburi’s Curry Puff, which is the medium-GI dessert, had a sugar content of 15.5 g/carbohydrate 50 g (sucrose 13.5 g/carbohydrate 50 g), and the group that consumed this had a sugar intake from the diet of 35.6 g, which was similar to the high-GI dessert group. The observed GI differences are probably due to the diverse ingredients and their ratios, particularly the types and amounts of carbohydrates and sugars used in these desserts [64]. According to Tily et al. [65], interactions with the fat and protein content in food, as well as other factors, can delay carbohydrate digestion, which affects carbohydrates consumed and explains the variation in the postprandial glycemic response. However, it may be because low- and medium-GI desserts have similar fats and proteins as well as dietary intake and possibly delay digestion compared to high-GI desserts, which in turn might alter the gut bacteria.

Our study observed a decrease in the abundance of the phylum Actinobacteriota in the high-GI dessert group compared to the low-GI and medium-GI dessert groups. Furthermore, the high-GI dessert group showed a tendency to have a decreased abundance of bacteria in the genus *Bifidobacterium*, as well as a significant decrease in *Collinsella* compared to the low-GI and medium-GI dessert groups. In addition, *Collinsella* showed a strong positive correlation with protein intake in the low-GI dessert group. The genus *Collinsella* is a member of the phylum Actinobacteria and the major taxon in the family Coriobacteriaceae. Members of this family are regarded as pathobionts. They may affect metabolism by altering the process of absorbing cholesterol in the intestines, reducing the production of glycogen in the liver, and enhancing the synthesis of triglycerides [66]. According to a number of studies, enteric infections’ virulence and pathogenicity may be altered by *Bifidobacterium* and *Collinsella* via altering the host bile acids [67]. A previous study showed that changes in the abundance of *Collinsella* bacteria may affect the level of cholesterol in the host’s blood plasma [68]. There is evidence that the abundance of *Collinsella* depends on the host’s nutritional intake, even when accounting for the considerable inter-individual variance in the gut microbiota in response to diet. For example, a high-protein weight-loss diet significantly reduces the amount of *Collinsella* bacteria [8], whereas consuming a Western diet pattern may lead to a higher abundance of several species of *Collinsella* [69]. More studies also revealed that low dietary fiber consumption may be associated with the abundance of *Collinsella* in the gut microbiome, highlighting the positive correlation between *Collinsella* prevalence and circulating insulin [70,71,72].

The genus *Bifidobacterium* (members of the phylum Actinobacteria), which is saccharolytic and produces acid but not gas from a variety of carbohydrates [73], is the dominant member of the human intestinal microbiota, although estimates of the bifidobacterial load in adults range from 4.4% [74] to 15% of the total fecal microbiota [75]. Bifidobacteria species are recognized as common gut bacteria that are believed to have beneficial effects on human health. Recently, a lack of them has been found to be associated with various disease conditions [76,77]. Moreover, the type of dietary carbohydrates may affect the abundance of specific species because different types of carbohydrates are sources of energy for specific bacteria [73]. As saccharolytic bacteria, *Bifidobacterium* digests a variety of carbohydrates to produce acid but not gas. Arabinoxylans from wheat and other grains are degraded by Bifidobacteria; therefore, individuals who consume a low-gluten diet, which reduces gluten-rich food items including wheat, barley, and rye cereals, have a reduced abundance of these bacteria within their intestinal tracts [78]. As mentioned above in our study, the main ingredient in Phetchaburi’s Custard Cake (low-GI dessert) was egg whites, sugar, and taro; Saraburi’s Curry Puff (medium-GI dessert) was wheat flour, potato, and onion; and Lampang’s Crispy Rice Cracker (high-GI dessert) was mainly sticky rice. The amount of dietary fiber in these desserts was 1.18 g, 2.79 g, and 0.96 g, respectively [79]. Moreover, during the intervention, the average dietary fiber intake of the low-GI, medium-GI, and high-GI dessert groups was 5.2 g, 6.1 g, and 5.7 g, respectively. There were no significant differences between the dessert groups. Mano et al. [16] investigated the influence of the main staple foods, white bread and white rice, on the gut microbiota using a commercially available package of side dishes. The results show that a 7-day intake of white bread containing a higher amount of dietary fiber than rice induced a higher abundance of the phylum Actinobacteria and the genus *Bifidobacterium*. Li et al. [15] showed that the dietary pattern of eating staple carbohydrates, including wheat, rice, and oats, for a week consecutively altered the structure of the intestinal microbial community. It was observed that wheat favored bifidobacteria, while rice suppressed bifidobacteria, and wheat suppressed the genera *Lactobacillus*, *Ruminococcus*, and *Bacteroides*. Hur et al. [54] studied the beneficial effects of a low-glycemic diet representing the traditional balanced Korean diet and a Westernized diet as a control diet. The low-glycemic diet was mainly composed of whole grains with fish, vegetables, seaweeds, and perilla oil, whereas the control diet contained refined rice, bread, noodles, meats, and processed foods. In fecal bacteria, *Bifidobacterium longum* and *Collinsella aerofaciens* were higher in the control diet group than in the low-glycemic-diet group. In contrast, Fava et al. [18] reported that the high-carbohydrate/high-glycemic diet increased the genus *Bifidobacterium* compared to the control group in subjects at risk of metabolic syndrome. Furthermore, a previous study showed that type 2 diabetes was positively associated with the genus *Ruminococcus*, *Fusobacterium* and *Blautia*, and negatively associated with *Bifidobacterium*, *Bacteroides*, *Faecalibacterium*, *Akkermansia*, and *Roseburia* [80,81]. Indeed, researchers found that *Lactobacillus*, *Bifidobacterium*, *Clostridium*, and *Akkermansia* are a group of bacteria that alter insulin resistance measures by reducing carbohydrate absorption, increasing energy use, making it easier for some intestinal enzymes to work, and being anti-inflammatory [56].

Based on the differential abundance analysis by LEfSe, we found that the gut microbiota was responsive to diet. In the high-GI dessert group, the genus *Klebsiella*, which belongs to the phylum Proteobacteria, was enriched after 24 h of consumption of Lampang’s Crispy Rice Cracker. *Klebsiella* bacteria are opportunistic pathogens commonly present in the flora of healthy individuals’ noses, throats, skin, and intestinal tracts. They can lead to various infections such as pneumonia, soft tissue and surgical wound infections, urinary tract infections, bloodstream infections, and sepsis [82]. Some studies have shown that consuming a diet high in sucrose can alter the functioning of *Lactobacillus plantarum* in the gastrointestinal system, resulting in sucrose-induced dysbiosis. This condition is characterized by an increase in the levels of *Clostridia* and *Bacilli* and a significant decrease in the levels of *Lactobacillus* spp., *Sphingomonas*, and *Klebsiella* [83,84]. Furthermore, in the low-GI dessert group, bacteria in the family Porphyromonadaceae and the genus *Porphyromonas*, which belong to the phylum Bacteroidota, were found to be the core microbiota after 24 h of consumption of Phetchaburi’s Custard Cake, which had the highest sugar content among the three Thai desserts tested here. These results indicated that after 24 h of intervention, the two Thai desserts, Lampang’s Crispy Rice Cracker and Phetchaburi’s Custard Cake, could contribute to the change in some core microbiota. Some common genera of anaerobic bacteria causing infectious diseases in humans include *Porphyromonas*, *Bacteroides*, *Prevotella*, *Fusobacterium* (Gram-negative bacilli), and *Clostridium* (Gram-positive bacilli), which belong to the phylum Bacteroidota [85]. A number of species in the genus *Porphyromonas* are classified as pathogens and are related to infections in humans or animals. Specifically, *P. gingivalis* has a significant role in causing periodontal disease [86]. A previous study has shown that the translocation of oral bacteria, such as *P. gingivalis*, to the gut can disrupt the balance of the gut microbiome [87]. Changes in the gut microbiota have been associated with the relationship between periodontal and systemic diseases [88]. *Porphyromonas* has been identified in the digestive tract, with two primary species: *P. asaccharolytica* [89] and *P. gingivalis* [90]. *Porphyromonas* had a high range of relative abundance (0.01–16%) and was identified in the distal mucosa of healthy subjects, specifically the species *P. asaccharolytica* [89]. Also, various bacteria, such as *P. gingivalis* and *P. asaccharolytica*, were found to have a significant association with colorectal cancer. In addition, they have been suggested as a new biomarker for the early detection of adenomatous polyps and colon cancer [91,92]. We found no significant difference between dessert groups in the abundance of the genera *Bacteroides*, *Prevotella*, *Fusobacterium*, and *Clostridium* (Appendix A). However, our results revealed that the abundance of the genus *Porphyromonas* in the low-GI dessert group (~0.1%) was higher than that in the medium-GI dessert group (0.001%) and the high-GI dessert group (0%) after 24 h of intervention (Appendix A).

Desserts, often consumed as snacks or with main meals, are significant but often overlooked dietary components [93]. Currently, snacking is a common part of people’s dietary habits. Previous research suggests that people worldwide obtain approximately 22% of their total daily calorie consumption from snacks [94,95]. Thai desserts, unlike Western-style desserts, do not contain milk or butter but are rich in flour, various carbohydrates (such as rice), sugar, and coconut milk. They are high in energy, fat, and carbohydrates and exhibit a range of GI values with uniformly high GL values [21]. Improper or excessive carbohydrate intake from these sources can lead to adverse health effects, including metabolic disorders [96]. Factors beyond the glycemic index, such as individual phenotypes and gut microbiota, also influence postprandial blood glucose responses and insulin control [97,98]. High-GI desserts are associated with an increased risk of metabolic syndrome [96].

Current treatments may not always provide adequate metabolic regulation. Probiotic therapy is gaining more attention as a supplementary strategy for preventing and treating chronic metabolic diseases [99]. A recent study found that *Lactiplantibacillus plantarum* dfa1, which was previously isolated from the Thai population, exhibited probiotic characteristics in vitro that may differ from the Caucasian probiotics and possessed effectiveness against lipid-induced intestinal damage [100]. The difference in such probiotic characteristics may be due to ethnicity, diet, environment, and community co-evolution [101,102]. According to Ondee et al. [13], an animal model fed a high-glucose or -fructose diet with or without *L. plantarum* dfa1 showed that both sugars induced similar obesity and prediabetes severity [100]. *L. plantarum* dfa1 administration reduced the risk of prediabetes and obesity risk, with slight gut microbiota alterations. Probiotics affected microbiome parameters, including the Chao1 index and abundance of Lactobacillus spp., potentially influencing insulin resistance [56]. An intriguing recent study, also in an animal model fed a high-fiber dessert, Prachuap Khiri Khan’s Pineapple Cheese Cake Biscuit, known for its high fiber content [103] and high glycemic index of 87.4 [21], with *L. plantarum* dfa1 supplementation showed a significant increase in *Akkermansia* and improved metabolic health [103]. This indicates that probiotics and dietary fiber work synergistically to enhance gut and metabolic health [103]. Therefore, using beneficial bacteria may be recommended to prevent metabolic disorders caused by carbohydrate consumption [13,103].

Our study’s randomized controlled design enhances the reliability of the findings, but limitations include the small sample size, short intervention period, and partial control over participants’ non-study meals. While a 24 h period may not fully capture longer-term impacts on the gut microbiota, it provides insight into immediate responses to GI variations. The lack of significant differences in alpha and beta diversity indices may reflect the limited timeframe. Nonetheless, the design approximates real-world dietary scenarios, allowing the exploration of how short-term dietary changes affect the microbiome [104].

Future studies should expand sample size, include longer follow-up periods, and impose stricter dietary control to better understand GI effects on the gut microbiota. Incorporating specific probiotic strains in these studies could yield insights into improving gut health through targeted dietary strategies. Additionally, studying personalized responses to dietary interventions could further elucidate diet–microbiome interactions, paving the way for individualized nutrition approaches.

## 5. Conclusions

This study examined the influence of Thai desserts with varying glycemic indices on the gut microbiome profiles of healthy participants over a 24 h period. We observe a trend in differences in the relative abundance of gut bacteria between dessert groups, although the change in bacterial abundance may be slight over a brief period. The consumption of high-glycemic index (GI) desserts resulted in a decrease in the relative abundance of Actinobacteria, particularly *Collinsella* and *Bifidobacterium*, and an increase in *Roseburia* and *Ruminococcus* compared to low- and medium-GI desserts. Additionally, there was a significant negative correlation between sugar intake and *Lactobacillus* abundance in the medium- and high-GI groups, as well as a moderately negative trend between sugar intake and *Akkermansia* abundance in the high-GI group.

## Figures and Tables

**Figure 1 nutrients-16-03933-f001:**
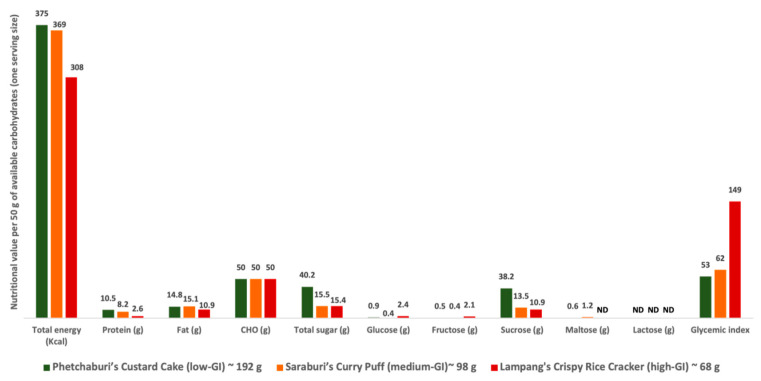
Nutritional value per 50 g of available carbohydrates (one serving size) of Thai desserts for testing. Note: Nutritional value data were retrieved from Namjud et al. [21], except for total sugar content, which was assessed in this study. The serving size was 50 g of available carbohydrates [26]. ND = not detected; GI = glycemic index; low GI ≤ 55, medium GI = 56–69, and high GI ≥ 70 [27].

**Figure 2 nutrients-16-03933-f002:**
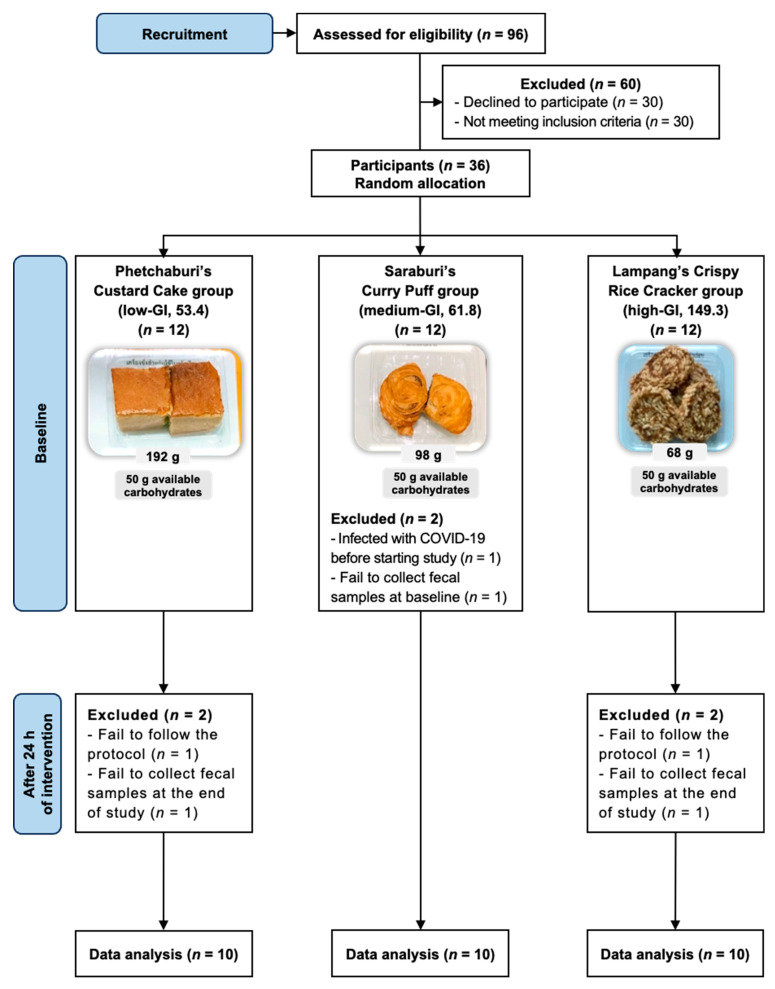
CONSORT flow chart of the experimental design in this study.

**Figure 3 nutrients-16-03933-f003:**
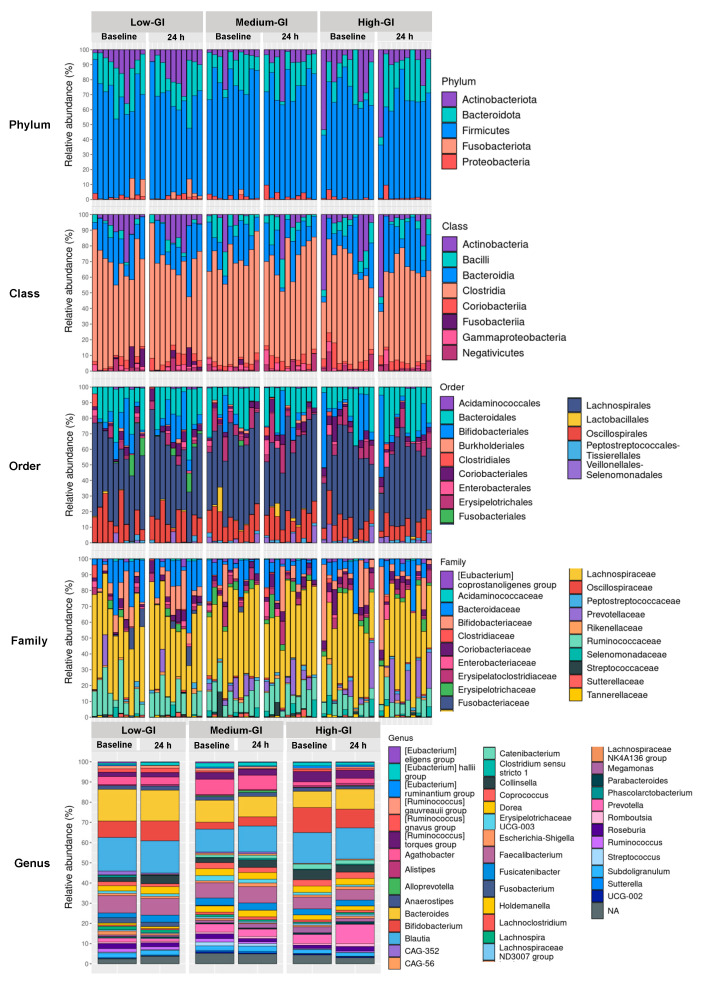
The relative abundance of the identified phyla, classes, orders, families, and genera of only the top 100 taxa. The relative abundance data of all identified taxa are presented in Appendix A.

**Figure 4 nutrients-16-03933-f004:**
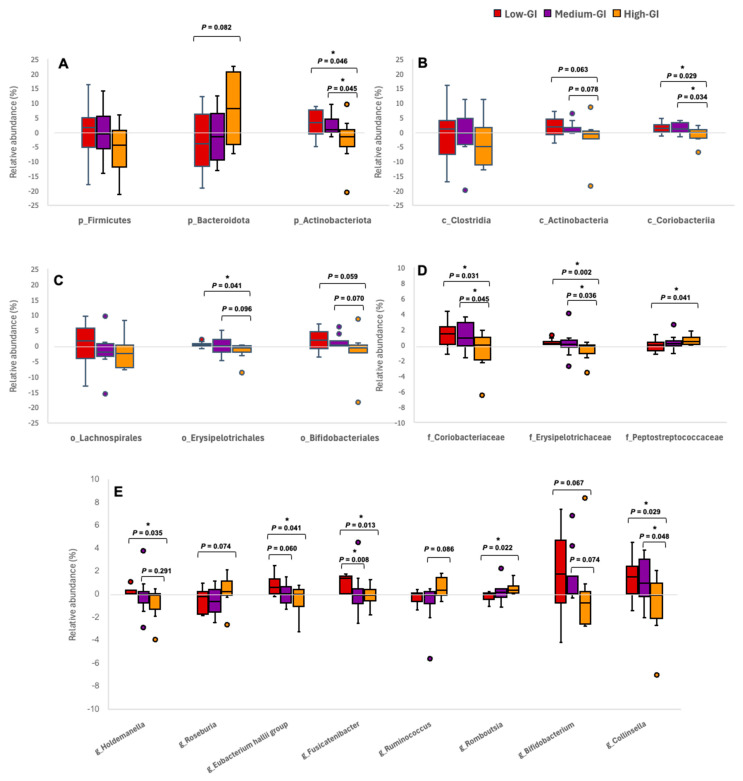
Boxplots showing the comparison of changes in relative abundance of the gut microbiota after the 24 h consumption of testing Thai desserts. The gut microbiome profiles were compared at the level of phylum (**A**), class (**B**), order (**C**), family (**D**), and genus (**E**). Data were analyzed using the Kruskal–Wallis test and Dunn’s post hoc test, with the Benjamini-Hochberg method. Asterisks (*) indicate significant differences at adjusted *p* < 0.05. More detailed data on changes in relative abundance are presented in Appendix A.

**Figure 5 nutrients-16-03933-f005:**
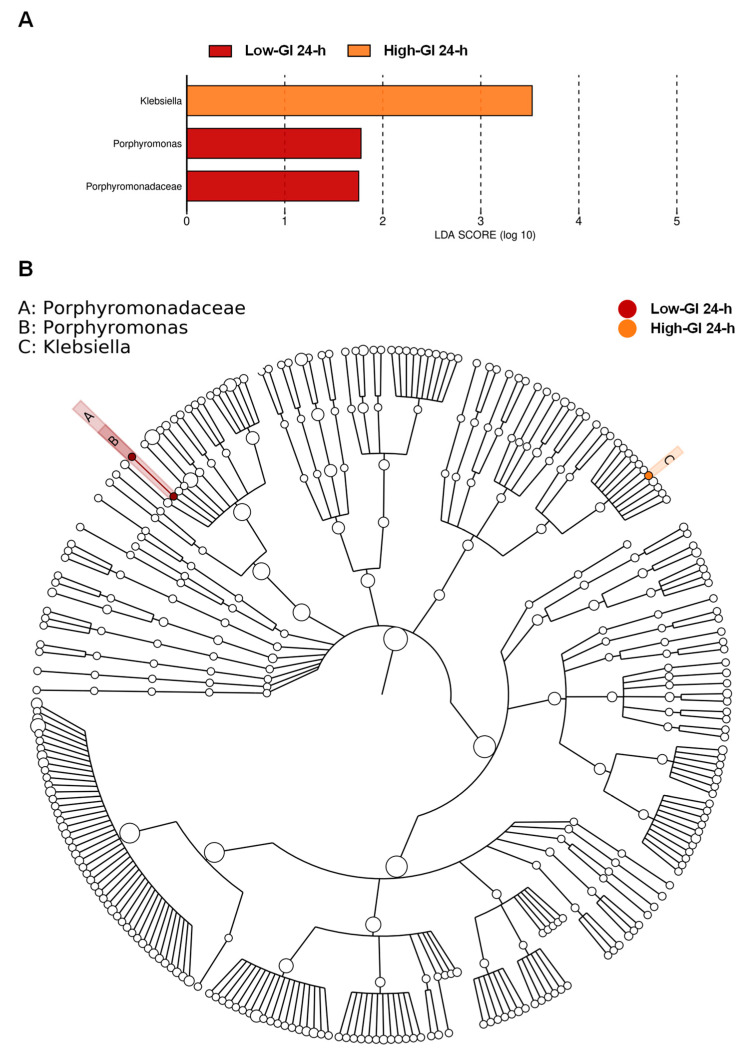
The gut microbial taxa determined as dominant biomarkers for association with the consumption of Thai desserts. After the 24 h intervention in the low-GI and high-GI dessert groups, the three taxa, the family Porphyromonadaceae, and the genera *Porphyromonas* and *Klebsiella* were determined as dominant biomarkers as analyzed by LEfSe with an LDA score ≥ 1.0 (**A**). Cladogram exhibiting the phylogenetic distributions of gut microbiota (**B**).

**Figure 6 nutrients-16-03933-f006:**
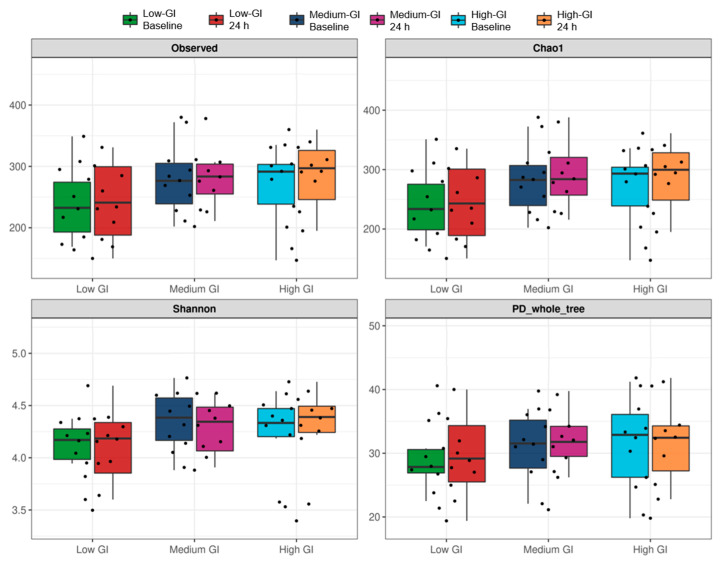
The alpha diversity indices of the dessert groups are shown by boxplots. The indices consist of observed ASVs, Chao1, Shannon, and PD whole tree. Black dots represent 20 samples obtained from combining the index values of both baseline and after the 24 h intervention for each dessert group. Alpha diversity values of each sample and quartiles of the distribution (minimum, first quartile, median, third quartile, and maximum of boxes) are demonstrated. No significant differences between dessert groups were observed by One-Way ANOVA or the Kruskal–Wallis test on any indices.

**Figure 7 nutrients-16-03933-f007:**
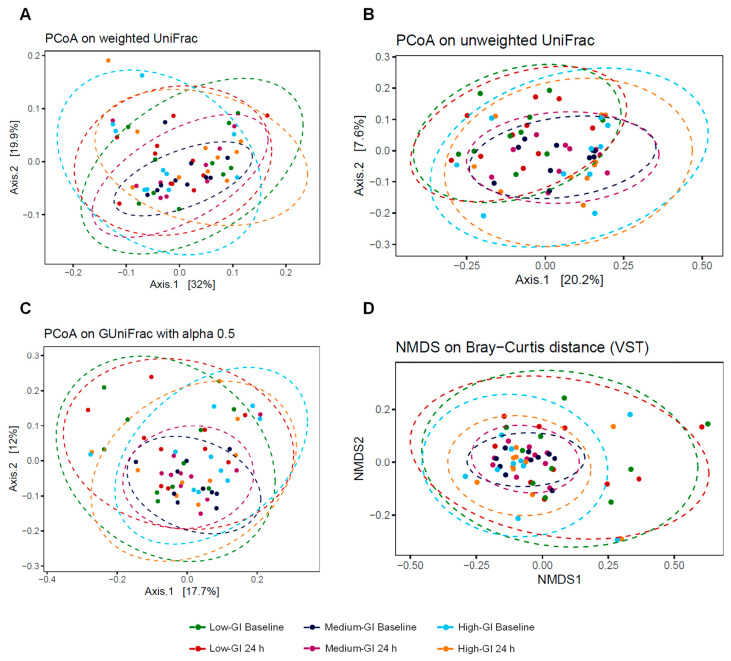
The beta diversity indices between dessert groups as shown by 2-dimentional plots. The indices consist of PCoA on weighted (**A**) and unweighted (**B**) UniFrac distances, or GUniFrac distances (**C**), and NMDS based on Bray–Curtis dissimilarity (**D**). Colored dots represent samples at baseline or after the 24 h intervention for each dessert group. No significant difference between dessert groups was observed by the PERMANOVA test in any indices.

**Figure 8 nutrients-16-03933-f008:**
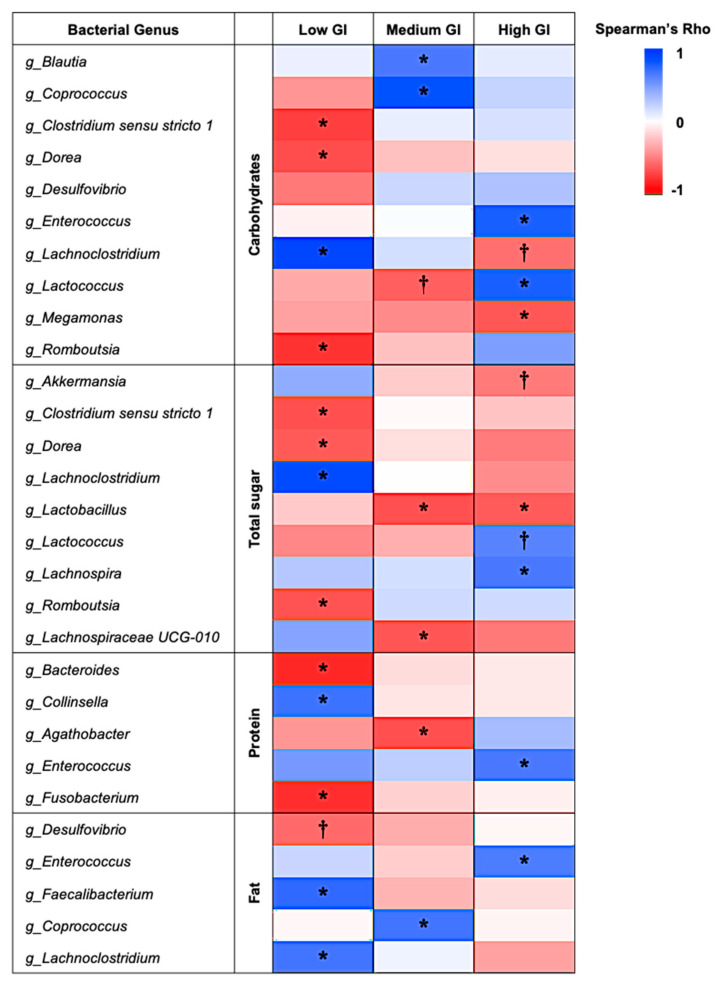
Correlation between dietary nutrient intake and the relative abundance of gut microbiota at the genus level. The correlation data were analyzed by Spearman’s rank correlation coefficient. Asterisks (*) indicate statistically significant Spearman’s rho (*ρ*) values at *p* < 0.05, while daggers (†) indicate the tendency of correlation at 0.05 ≤ *p* < 0.10.

**Table 1 nutrients-16-03933-t001:** Characteristics of the participants at the baseline.

Characteristics	Tested Thai Desserts		
	Phetchaburi’s Custard Cake (Low-GI, *n* = 10)	Saraburi’s Curry Puff (Medium-GI, *n* = 10)	Lampang’s Crispy Rice Cracker (High-GI, *n* = 10)
Gender			
Male (number)	2 (20)	2 (20)	1 (10)
Female (number)	8 (80)	8 (80)	9 (90)
Age (years)	29.2 ± 7.3	30.7 ± 7.8	28.2 ± 7.2
Anthropometrics			
Body weight (kg)	56.1 ± 6.8	55.8 ± 4.3	50.8 ± 6.7
Height (cm)	164.0 ± 6.7	163.4 ± 6.3	157.9 ± 5.9
BMI (kg/m^2^)	20.8 ± 1.4	20.9 ± 1.0	20.3 ± 1.5
Waist circumference (cm)	72.2 ± 5.3	73.1 ± 6.0	69.6 ± 5.7
Vital signs			
Systolic blood pressure (mmHg)	110 ± 10.9	110.9 ± 12.5	107.9 ± 13.7
Diastolic blood pressure (mmHg)	66.1 ± 6.9	65.9 ± 7.5	67.1 ± 10.0
Pulse (bpm)	78.4 ± 11.8	85.5 ± 13.2	83.6 ± 14.7
Blood biochemical parameters			
Fasting blood sugar (mg/dL)	83.7 ± 6.6	83.9 ± 5.9	85.6 ± 5.8
Hemoglobin A1c (%)	5.0 ± 0.4	5.0 ± 0.3	5.1 ± 0.3
Total cholesterol (mg/dL)	180.6 ± 25.6	169.9 ± 17.8	176.2 ± 25.9
Triglycerides (mg/dL)	48.0 (42.0, 63.0)	70.5 (45.0, 79.0)	55.5 (41.0, 68.0)
HDL-C (mg/dL)	62.2 ± 10.0	60.8 ± 14.8	63.5 ± 14.3
LDL-C (mg/dL)	108.4 ± 33.4	97.7 ± 26.6	106.4 ± 29.0
Blood urea nitrogen (mg/dL)	9.8 (9.0, 10.1)	11.0 (9.9, 11.9)	9.3 (8.0, 10.5)
Creatinine (mg/dL)	0.7 (0.6, 0.8)	0.7 (0.6, 0.8)	0.7 (0.6, 0.9)
Alanine aminotransferase activity (IU/L)	15.0 (12.0, 17.0)	14.0 (11.0, 17.0)	11.5 (10.0, 14.0)
Aspartate aminotransferase activity (IU/L)	17.4 ± 3.3	18.0 ± 2.1	15.5 ± 4.0
Duration of fecal sample collection after intervention (h)	24.8 ± 3.1	23.8 ± 3.0	23.7 ± 2.1

Note: Data are presented as number (%), mean ± SD, or median (25th, 75th percentile). There were no significant differences in any characteristics between the testing dessert groups as analyzed by Fisher’s exact test, One-Way ANOVA, or the Kruskal–Wallis test.

**Table 2 nutrients-16-03933-t002:** Energy intake and dietary composition at baseline and after 24 h intervention.

Energy Intake and Dietary Composition	Tested Thai Desserts		
	Phetchaburi’s Custard Cake (Low-GI, *n* = 10)	Saraburi’s Curry Puff (Medium-GI, *n* = 10)	Lampang’s Crispy Rice Cracker (High-GI, *n* = 10)
**At baseline**			
Total energy (kcal/day)	1172.9 ± 252.4	1151.3 ± 250.8	1049.3 ± 232.8
Energy from carbohydrates (%)	43.3 ± 5.5	40.7 ± 8.5	45.8 ± 7.5
Energy from protein (%)	22.5 (19.8, 25.0)	18.5 (17.9, 19.8)	19.3 (16.0, 21.8)
Energy from fat (%)	34.7 ± 3.4	39.5 ± 7.9	35.2 ± 5.3
Carbohydrates (g)	126.0 ± 26.9	118.6 ± 43.0	121.5 ± 37.6
Total sugar (g/day)	31.3 ± 19.3	23.2 ± 16.4	26.9 ± 16.7
Protein (g/day)	65.2 ± 20.4	56.6 ± 13.8	49.3 ± 13.2
Fat (g/day)	45.4 ± 10.8	50.1 ± 13.5	40.7 ± 9.7
Dietary fiber (g/day)	7.7 (5.5, 10.9)	5.3 (4.9, 8.4)	5.3 (4.5, 7.3)
**After 24 h intervention**			
Total energy (kcal/day)	1425.0 ± 457.5	1302.0 ± 337.3	1503.6 ± 378.4
Energy from carbohydrates (%)	42.0 ± 7.1	42.6 ± 6.9	46.9 ± 3.4
Energy from protein (%)	17.1 (15.3, 26.5)	18.4 (17.1, 21.8)	16.6 (13.4, 16.8)
Energy from fat (%)	38.2 ± 4.0	39.1 ± 7.1	37.5 ± 2.5
Carbohydrates (g)	148.1 ± 46.1	136.3 ± 30.2	175.8 ± 44.7
Total sugar (g/day)	56.9 (43.5, 62.2) ^a^	25.6 (20.8, 41.8) ^b^	27.7 (22.9, 33.8) ^b^
Protein (g/day)	65.4 (42.6, 79.0)	59.2 (44.8, 70.9)	61.4 (50.2, 68.0)
Fat (g/day)	59.6 ± 15.7	57.5 ± 21.0	62.7 ± 16.5
Dietary fiber (g/day)	5.2 ± 3.1	6.1 ± 2.4	5.7 ± 2.3

Note: Data are presented as mean ± SD or median (25th, 75th percentile). There were no significant differences in energy intake and dietary composition, except total sugar intake after the 24 h intervention. ^a, b^ Different superscripts in the same row indicate significant differences between groups using One-Way ANOVA with the Bonferroni post hoc test (adjusted *p*-value < 0.05) or Kruskal–Wallis and Dunn post hoc tests for multiple comparisons, with the Benjamini–Hochberg method (adjusted *p*-value < 0.05).

## Data Availability

The sequence reads of 16S rRNA genes were submitted to the GenBank database and are accessible online with the Sequence Read Archive (SRA) accession number SRS15280189 (https://www.ncbi.nlm.nih.gov/sra/?term=SRS15280189, accessed on 1 October 2023).

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
