# Peer review of "Sugar Composition of Thai Desserts and Their Impact on the Gut Microbiome in Healthy Volunteers: A Randomized Controlled Trial"

_nutrients, 2024, doi:10.3390/nu16223933_

Round 1

Reviewer 1 Report

Comments and Suggestions for Authors

This study provides a preliminary glimpse into the gut microbiome's response to carbohydrate-rich Thai desserts, contributing to the broader understanding of diet-microbiome interactions. However, the small sample size, brief intervention window, and lack of detailed quantitative results weaken the impact of the findings. To advance this line of research, future studies should employ larger, more diverse populations, consider more controlled food interventions, and integrate longer follow-up periods to determine the true implications of glycemic index variations on gut health and metabolic diseases.

Comments on the Quality of English Language

Can be improved.

Author Response

Reviewer 1: This study provides a preliminary glimpse into the gut microbiome's response to carbohydrate-rich Thai desserts, contributing to the broader understanding of diet-microbiome interactions. However, the small sample size, brief intervention window, and lack of detailed quantitative results weaken the impact of the findings. To advance this line of research, future studies should employ larger, more diverse populations, consider more controlled food interventions, and integrate longer follow-up periods to determine the true implications of glycemic index variations on gut health and metabolic diseases.

Response: Thank you for your insightful feedback. We acknowledge the limitations of our study, including the small sample size, short intervention period, and preliminary nature of the findings. In response, we have clearly highlighted these limitations in the revised Discussion section, emphasizing that this study serves as an initial exploration of the impact of Thai desserts with varying glycemic indices on gut microbiota composition. We also discuss how future research should incorporate larger, more diverse populations, extended follow-up periods, and more controlled dietary interventions to comprehensively assess the long-term implications of glycemic index variations on gut health and metabolic outcomes. Your suggestions have been invaluable in refining our study’s presentation and future directions, and we appreciate your guidance in strengthening this line of inquiry.

Reviewer 2 Report

Comments and Suggestions for Authors

The article presented on the Sugar Composition of Thai Desserts and Their Impact on Gut Microbiome is well presented, well done and also provides novel comments on the glycemic index and its relationship with the microbiota, which is why it is of interest to Thai nutritionists and those from other parts of the world where there are Thai restaurants. It therefore deserves to be published.

Some modifications would improve the reading by interested nutritionists if superfluous sections and decimals were eliminated.

- In the abstract, eliminate the paragraphs between lines 20-24 and 42-44. The first is superfluous and the second, which refers to future studies, is displayed at the end of the discussion.

- In the introduction, eliminate the paragraphs between lines 65-73, 94-102, 130-134, because they are superfluous.

- In the conclusions, delete the paragraphs between lines 856 and 860, future studies are not conclusions and are already presented at the end of the discussion.

- Delete the decimals of the Kcal values ​​because they are not significant for the diets, for example in Fig 1.

- Delete the decimals of the GI values ​​because they are not significant, for example in lines 220, 225, 229, 347, 349, Table 3, and others.

Author Response

Reviewer 2: The article presented on the Sugar Composition of Thai Desserts and Their Impact on Gut Microbiome is well presented, well done and also provides novel comments on the glycemic index and its relationship with the microbiota, which is why it is of interest to Thai nutritionists and those from other parts of the world where there are Thai restaurants. It therefore deserves to be published.

Response: Thank you for your positive feedback and support for our study’s publication. We are glad that you found the insights on the glycemic index of Thai desserts and their impact on the gut microbiome to be relevant and valuable for nutritionists in Thailand and globally. We appreciate your encouraging remarks.

Reviewer 2: Some modifications would improve the reading by interested nutritionists if superfluous sections and decimals were eliminated.

Response: Thank you for your suggestion. We have removed superfluous sections and unnecessary decimals to improve readability for nutritionists and other interested readers.

Reviewer 2: In the abstract, eliminate the paragraphs between lines 20-24 and 42-44. The first is superfluous and the second, which refers to future studies, is displayed at the end of the discussion.

Response: The Abstract was revised accordingly.

Reviewer 2: In the introduction, eliminate the paragraphs between lines 65-73, 94-102, 130-134, because they are superfluous.

Response: The Introduction was revised as suggested.

Reviewer 2: In the conclusions, delete the paragraphs between lines 856 and 860, future studies are not conclusions and are already presented at the end of the discussion.

Response: The Conclusion was revised accordingly.

Reviewer 2: Delete the decimals of the Kcal values because they are not significant for the diets, for example in Fig 1.

Response: The Kcal values were corrected as advised.

Reviewer 2: Delete the decimals of the GI values because they are not significant, for example in lines 220, 225, 229, 347, 349, Table 3, and others.

Response: The GI values were corrected as appropriate.

Reviewer 3 Report

Comments and Suggestions for Authors

This is an interesting study concerning how different desserts affect gut microbiome. Introduction justifies the study, methods which were applied are appropriate, results are almost clearly presented (Figure 3. is barely readable, increase the font in the legend) and very well discussed. Authors indicate some strength and limitations of their study as well as future needs. 

I have two serious remarks. 

1) As your diet intervention lasted only one day, it is worth mention it as a study limitation. If you study lasted at least one week the strength of the study would be much better.

2) Where does the information on the glycemic index of the tested desserts come from? Give this information in materials and methods chapter.

Besides:

Use italics for all microbiological names, e.g. line 57, 86, 175, 469, and many more

Sometimes italics are not necessary, line 449, the word "group"

Author Response

Reviewer 3: This is an interesting study concerning how different desserts affect gut microbiome. Introduction justifies the study, methods which were applied are appropriate, results are almost clearly presented (Figure 3. is barely readable, increase the font in the legend) and very well discussed. Authors indicate some strength and limitations of their study as well as future needs.

Response: Thank you for your positive feedback and for highlighting areas for improvement. We have increased the font size in the legend of Figure 3 to improve readability as suggested.

Reviewer 3: 1) As your diet intervention lasted only one day, it is worth mention it as a study limitation. If you study lasted at least one week the strength of the study would be much better.

Response: Thank you for this valuable suggestion. We acknowledge that the one-day intervention is a limitation, as it may not capture long-term effects on the gut microbiome. We have clarified this limitation in the Discussion section, noting that a longer intervention period, such as one week, would strengthen the study’s findings and provide a more comprehensive understanding of the impact of glycemic index variations on gut health.

Reviewer 3: 2) Where does the information on the glycemic index of the tested desserts come from? Give this information in materials and methods chapter.

Response: Thank you for your question. The glycemic index information for the tested desserts has just been published, and we have updated Reference #34 accordingly.

Reviewer 3: Use italics for all microbiological names, e.g. line 57, 86, 175, 469, and many more Sometimes italics are not necessary, line 449, the word "group".

Response: The microbiological names were corrected as advised.

Round 2

Reviewer 1 Report

Comments and Suggestions for Authors

The manuscript was improved.